# Productive Performance and Carcass Characteristics of Broiler Chickens Fed on Diets with Different Protein, Energy Levels, and Essential Oils During the Warm Season in Dry Tropics

**DOI:** 10.3390/ani14223179

**Published:** 2024-11-06

**Authors:** Jorge H. García-García, Jaime Salinas-Chavira, Flaviano Benavides-González, Enrique Corona-Barrera, Elvia M. Romero-Treviño, Jorge Loredo-Osti

**Affiliations:** 1College of Veterinary Medicine and Animal Science, Autonomous University of Tamaulipas, Victoria 87000, Mexico; a2213390018@alumnos.uat.edu.mx (J.H.G.-G.); jsalinas@docentes.uat.edu.mx (J.S.-C.); flbenavides@docentes.uat.edu.mx (F.B.-G.); enrique.corona@docentes.uat.edu.mx (E.C.-B.); 2Altamira Technological Institute, National Technological Institute of Mexico, Altamira 89600, Mexico; elvia.rt@altamira.tecnm.mx

**Keywords:** dry tropics, broiler, oregano, cinnamaldehyde, diets

## Abstract

In tropical areas, the productive performance of broiler chickens is reduced during the summer by the adverse effects of climatic conditions. Essential oils (EOs) of plants added in the feed have contributed to alleviate heat stress in broiler chickens. The negative impact of heat stress has also been reduced through feed restriction. This study was conducted in a tropical area in the hot season with two hundred, 1-day-old chickens fed diets with or without EOs, combined with standard or reduced nutrient diets. Results revealed greater weight gain or carcass weight of broiler chickens fed standard diets than the chicks fed the reduced nutrient diets. Essential oil supplementation tended to improve feed conversion efficiency in the chicks. Carcass traits of chicks were not influenced by EO additions to the diets. Body temperature was lower in chicks receiving diets with low nutrient concentration, as well as those receiving EOs. In total, broiler chickens in this study were under heat stress and both strategies, nutrient reduction and EO addition to diets, contributed to ameliorate body temperature or feed conversion of the chicks. Greater weight gain or carcass traits were found for chicks fed standard diets, though these had a negative impact on the mortality of the birds.

## 1. Introduction

Broiler chickens are commercially important and provide a key nutrient source for the population. In Mexico, by the year 2021, the chicken industry accounted for 34.2% of the total livestock production. This kind of meat is the main source of protein for the Mexican population, with a consumption of 33.47 kg/year per capita. Of the total consumption of chicken meat, about 85.82% of chicken meat was produced in Mexico (3,664,968 t), and the rest was imported (605,418 t). With these figures, Mexico is among the main consumers and producers of chicken meat in the world. It is highlighted that in Mexico, most of the chicken production is in areas of the country with warm climates that include tropical and subtropical types of weather, as well as other localities with warm weather conditions [1].

The animals are in heat stress when they produce more heat than their capability to dissipate it into the environment. Particularly, broiler chickens are sensitive to heat stress as their rapid weight gain infers high rates of metabolism with high heat increment, high body temperature, and a small zone of comfort for temperature (close to 20 °C). In addition, broiler chickens have limited capability to dissipate heat as feathers cover their body, they do not have sweat glands, and chickens use panting to dissipate heat. However, panting is limited at high relative humidity in the environment. Also, the high population density in intensive production farms contributes to the heat stress of broiler chickens [2,3].

The production and the well-being of broiler chickens under heat stress are impaired by mitochondrial oxidative stress where free radicals are released, increasing corticosterone that alters liver metabolism of lipids and cell death occurs [4,5]. Oxidative stress also affects the process of digestion and absorption of nutrients of birds under heat stress in various aspects, for example, a reduction in the area of intestinal villus has been observed, as well as in the transporters required for nutrient absorption and in intestinal enzymes [6,7,8]. Besides, the birds exposed to heat stress have increased peripheral blood circulation and reduced visceral circulation. The net result is a reduction in feed efficiency [7]. In total, heat stress in broiler chickens alters their physiology and metabolism, depresses the immune system, reduces feed intake and weight gain, alters the meat quality, and, in severe cases, increases mortality; as a result, heat stress inflicts enormous economic losses on the poultry industry [4,7,9,10].

Nutritional strategies that may contribute to alleviate the heat stress in broiler chickens include feed restriction and supplementation with phytochemicals (phytomolecules) [4,11]. The latter contain secondary metabolites of plants as active ingredients, and include different extracts, essential oils, and pure compounds that were isolated [4]. Phytochemicals have improved the physiology and metabolism of broiler chickens in different ways. They have reduced the oxidative stress by reducing free radicals from the metabolism; moreover, phytomolecules have improved the digestibility of nutrients, reduced body temperatures, reduced potential pathogenic gut microorganisms, and have also improved the immune system, among other variables that result in the improvement of performance of broiler chickens [12,13,14]. Phytochemicals have been proposed to contribute to mitigation of heat stress in broiler chickens [3,7,10,15]; however, other holistic management practices to decrease heat stress in the broiler chicken production should be considered together with the phytochemicals [4,8].

Feed restriction is postulated to reduce the negative effects of heat stress in broiler chicken production. There are two ways to restrict the feed. One is to withdraw feed during the peak hours of ambient temperature; the second one is a dual feeding, that is, feed the chickens with a standard diet during the cool hours, and thereafter feed the chickens with a diet reduced in nutrients (particularly crude protein) during the peak of high ambient temperature. These alternatives reduce the heat stress in the broiler chicken production. However, it is not used in commercial production because of the low applicability in the feeding management, and the animals require more days to fatten and reach market weight [4,8].

To the best of our knowledge, there is limited information on dietary nutrient reduction and oregano and cinnamon essential oils to alleviate, to some extent, the heat stress in broiler chickens. Therefore, the objective of the present study was to evaluate the effect of nutrient concentration in diets with the addition of essential oils from oregano and cinnamon on the productive performance and carcass characteristics of broiler chickens fed on diets with different protein and energy levels, and the essential oils, during the warm season in a dry tropical climate (when high ambient temperatures cause heat stress in the birds).

## 2. Materials and Methods

### 2.1. Area of Study

This study was carried out at the poultry farm of the College of Veterinary Medicine and Animal Science of the Autonomous University of Tamaulipas, located in Ciudad Victoria, Tamaulipas (a subtropical area in northeastern Mexico). The site is located at 23°44′06″ N and 97°09′50″ W, at an altitude of 323 m. The mean annual rainfall is 926 mm, and the average yearly temperature is 24 °C [16]. These climatic conditions are typical for the dry subtropics (ACw).

### 2.2. Animals and Diets

Two hundred (100 male, 100 female), 1-day-old, Ross 308 broiler chickens, weighing 44.5 ± 1.14 g (mean ± SD), were obtained from a commercial hatchery. The experiment started on June 29 and ended on August 10 of 2023. During this period, the average temperature and relative humidity were 32.0 ± 1.12 °C and 56.9 ± 5.20%, respectively. Temperature and relative humidity were recorded inside the coop, as it is described in Section 2.3 of the manuscript. The coop had windows along two opposite sides, and it was equipped with ventilation and heating systems. In the first week, the heating systems started functioning to maintain the room temperature at 33 °C, and then was reduced by 3 °C each week until the third week. In the second week, the ventilation was created only by opening the windows during the day, when temperatures were ≥30 °C. From the third week to the end of the study, when temperatures were ≥30 °C, the ventilation during the day was created by opening the windows plus the use of three industrial-type fans, usually from 10:00 am to 6:00 pm. Each treatment (diet) included 50 birds (25 males, 25 females) randomly assigned to five replicates of ten animals (5 males, 5 females) each. During the whole experiment, the birds were housed in 20 pens of concrete floor with grass straw (chopped into 2.5 cm particles) as bedding material. Twenty four hours of light per day was provided during the whole trial. Each pen had an automatic drinker and a manually filled feeder. Pen density was set to 10 birds per square meter and water and feed were offered ad libitum. Birds were vaccinated on day 7 of the trial against fowlpox (wing puncture) with a homologous strain.

The chickens were raised according to standard commercial practices. Two feeding phases were used: 1 to 21 (starter) and 22 to 42 days of age (finisher) (final weight 2563.2 ± 269.5 g). There were four treatments (T) for starter and finisher diets. This study considered two diets with different nutrient compositions, a standard diet (STD) and a nutrient reduced diet (RED). This study also considered the addition of essential oils (EOs) in diets at two different levels (0 ppm and 200 ppm). The experiment design considered four experimental treatment (T) diets: T1 (STD + 0 ppm EOs); T2 (STD + 200 ppm EOs); T3 (RED + 0 ppm EOs); and T4 (RED + 200 ppm EOs), with five replicates each. The essential oils were basically a combination of cinnamaldehyde and carvacrol (EW Nutrition, distributed in Mexico), which is a blended micro-encapsulated compound. The selection of 200 ppm of EOs was based on the technical note of the EO manufacturer, as by previous reports using similar dose of EOs [17,18]. The EOs in the diets considered that 200 ppm corresponds to 0.2 g EO/kg of feed, i.e., for 100 kg of feed there was 20 g of EO included (0.2 × 100 = 20). The EOs were first mixed with the premix and thus, the premix plus the EOs were mixed with the other ingredients of the diets, according to the respective treatments. The premix in the diets contained minerals including calcium and phosphorus, in addition to other minerals, amino acids, and vitamins. The RED diet had 10% less protein and 10% less metabolizable energy than the STD diet. Diets were prepared according to the National Research Council (NRC, 1994) for poultry recommendations as well as previous reports [19,20] (Table 1 and Table 2). Diets were analyzed for crude protein (CP) using the Kjeldahl-N method, following the procedures published by the AOAC (2006) [21]. Dry matter (DM) was determined by dehydration of 2 g of sample using an air-forced oven (Memmert, UFB 400, Schwabach, Germany) at 103 °C for 24 h. As observed (Table 1 and Table 2), the determined CP values exceeded the calculated ones by about 2%; this could be due to the CP contained in the premix. In the formulation we did not include CP in the premix because the technical recommendation does not consider CP; however, considering the amino acids and other additives, the premix must contain crude protein. It is recommended to analyze all ingredients of the diet, including the premix.

### 2.3. Temperature and Relative Humidity

Ambient temperatures and relative humidity were recorded across the study using four digital hygrometers (HTC-2, China). The devices were installed into four pens, distributed in four different locations. The measurements were taken at the height of the chickens, according to their age. The readings were taken daily at 09:00 h, corresponding to minimum and maximum temperatures (°C) at night and day, respectively. The temperatures corresponded to the previous day as the memory of the device had the registered values saved, so after taking the readings the device was reset to start a new recording.

Temperature humidity index (THI) was estimated using the next equation [22]:THI = Tdb − [(0.31 − 0.31 RH)(Tdb − 14.4)], 
where, THI = temperature humidity index, Tdb = ambient temperature (°C), and RH = relative humidity (%)/100.

Body temperature of chickens was recorded throughout the study using a thermographic camara (NF-521, China). The records included data of at least four chickens per cage. The approximate distance from the camera to the chickens was 1.0 m, and not more than 1.5 m. Body temperature measurements were recorded daily at 08:00, 12:00, 16:00, and 20:00 h in all cages.

### 2.4. Productive Performance

Body weight and feed intake were measured weekly, then the feed conversion ratio (FCR; feed intake, g/weight gain, g) was calculated. At the end of the feeding trial, two male chickens per cage were selected to be euthanized by cervical dislocation according to the Official Mexican Act [23] for carcass traits determination. The selection of the chickens was made by visual appreciation, considering birds within the average size in each cage. Birds of biggest or smaller sizes were excluded. Carcass weight without viscera was used to estimate hot carcass yield (carcass weight, g/live weight, g), right after the carcass was processed for major cuts: breast, thighs and drumsticks, wings, and back.

At the slaughter of chickens, after scalding, defeathering, and evisceration, the temperature and pH of the meat (pectoralis muscle) was recorded using a punch m thermometer (TP101, SAI Factory, Shanghai, China) and a pH tester (HI981036, Hanna Instruments, Woonsocket, RI, USA).

### 2.5. Mortality

The death of animals was recorded daily throughout the study. Mortality was determined by the proportion of the chickens that died and the chickens at the beginning of the study. Mortality was estimated in all treatments of the study. Mortality is presented in the results as descriptive statistics. The cero mortality in two treatments does not allow statistical analyses for analysis of variance and mean comparations between treatments.

### 2.6. Statistical Analyses

The data obtained were analyzed using a completely randomized design with a 2 × 2 factorial arrangement. The model assumptions for all variables were verified using the Shapiro–Wilk test for normality (*p* ≥ 0.18) and the Bartlett test for homogeneity of variances (*p* ≥ 0.11). The main effects were two diets with different nutrient concentrations (STD and RED) and addition of two EO levels (0 and 200 ppm), as well as the interaction between these effects.

For the productive performance (weight gain, feed intake, and FCR), and temperatures, each replicate was the average of broiler chickens per pen. For carcass evaluation, the replicate was the average of two birds (selected at random) per pen. Significance was set at *p* < 0.05, and tendency was set at *p* ≥ 0.05 and *p* ≤ 0.10. All the statistical analyses were performed on the generalized linear model procedure of SAS (Statistical Analysis System).

## 3. Results

### 3.1. Climatic Conditions

Average values of ambient temperature, relative humidity, and average body temperature of broiler chickens across the feeding study are shown in Figure 1 and Figure 2. As it is observed, the changes in ambient temperature had similar trends to changes in body temperature, although relative humidity exhibited a different pattern. Average ambient temperature (°C) was 34.2 ± 3.0, 33.1 ± 3.4, and 33.6 ± 3.2; relative humidity (%) was 45.2 ± 12.4, 49.3 ± 16.5, and 47.2 ± 14.7; temperature humidity index (THI) was 30.7 ± 1.8, 37.8 ± 3.1, and 34.1 ± 4.4; respectively in all cases for starter (1–21 d), finisher (21–42 d), and the total feeding period (1–42 d). Lower ambient temperatures were recorded during the finisher phase than during the starter phase, therefore the finisher phase had greater THI, as higher values of relative humidity were recorded. Figure 3 shows the THI across the study.

### 3.2. Productive Performance

Productive performances of chickens fed on diets with different nutrient concentrations and addition of EOs are shown in Table 3. In the starter phase, dietary treatments did not show an effect on feed intake (*p* ≥ 0.21). Weight gain was greater (*p* < 0.01, main effect) in chickens fed with the STD diets than in those fed with diets of low nutrient concentration. The average weight gain of chickens was 927 g and 807 g for the STD and low nutrient diets, respectively. The FCR (*p* = 0.09) tended to be better in chickens fed on the STD diets as compared to birds fed on the diets with low nutrient concentration. The addition of EOs did not show an effect (*p* ≥ 0.14) on feed intake, weight gain, or FCR of chickens.

In the finisher phase, greater feed intake (*p* = 0.02, main effect) and weight gain (*p* = 0.04, main effect) were observed in birds fed on the STD diets than in those fed on diets with low nutrient concentration. Average feed intake was 3328 g and 2999 g, and average weight gain was 1743 g and 1562 g, respectively for chickens fed on the STD diets and low nutrient diets. The FCR was similar (*p* = 0.97) between chickens receiving the STD diets and those fed on RED diets. The EO supplementation did not have an effect (*p* ≥ 0.14) on feed intake, weight gain, or the FCR.

Overall, greater feed intake (*p* = 0.02, main effect) and weight gain (*p* < 0.01, main effect) were observed in chickens fed on the STD diets than in those receiving diets with low nutrient concentration. Average feed intake was 4665 g and 4264 g, and average weight gain was 2670 g and 2368 g, respectively for the broiler chickens fed on the STD and the RED diets. The FCR ratio was similar (*p* = 0.51) between animals on the STD diets and animals on diets with low nutrients concentration.

The diets with the addition of EOs had no effect on the feed intake nor weight gain (*p* ≥ 0.26), however the FCR tended (*p* ≥ 0.08) to be better in chickens with the addition of EOs.

### 3.3. Carcass Evaluations

Results of carcass evaluations of broiler chickens fed on diets with different nutrient concentrations and addition of EO are shown in Table 4. The nutrient concentration did not influence (*p* ≥ 0.14) the values of carcass yield, breast yield, pH, or temperature of the meat. Hot carcass weight was greater (*p* < 0.01, main effect) in chickens fed on the STD diets than in those receiving the RED diets. Average of hot carcass weights were 3173 g and 2718 g for chickens fed on the STD and the RED diets, respectively.

Chickens fed on diets with low nutrient concentration had greater yields of leg-thigh (*p* = 0.01, main effect), back (*p* = 0.01, main effect), and wings (*p* < 0.01, main effect) as compared to the other groups. Average yield values of leg-thigh were 26.5% and 28.0%, back 20.7% and 21.9%, and wings 11.7% and 12.8%, for chickens fed on the STD and the RED diets, respectively. The addition of EOs did not influence (*p* ≥ 0.11) the carcass characteristics.

### 3.4. Body Temperature

The average body temperature measured with a thermographic camera, according to the feeding phase of chickens in the study, is shown in Table 5. There was no effect of treatment on body temperature of chickens during the starter phase (*p* ≥ 0.17) nor in the finisher phase (*p* = 0.08). The average body temperature (°C) in the finisher phase was 37.3 vs. 37.05, respectively for animals on the STD and the RED diets (Figure 4). During the length of the study (1–42 d), chickens fed on the RED diets showed lower (*p* = 0.03) body temperature than those birds fed on the STD diets. Average body temperature (°C) was 37.8 vs. 37.65, respectively for animals on the STD and reduced nutrients diets (Figure 5). Body temperature of chickens fed with the STD or the RED diets across the study are shown in Figure 6.

The addition of EOs had no effect (*p* ≥ 0.22) on the body temperature of the chickens. Body temperature of broiler chickens fed with EO diets across the study is shown in Figure 7. There was no interaction effect (*p* ≥ 0.66) of the type of diet and the addition of EOs on the body temperature of the chickens.

The body temperatures of broiler chickens, by feeding phase at different hours of the day, are shown in Table 6. There was no effect from treatment (*p* ≥ 0.12) on the body temperature measured in the starter phase. In the finisher phase of broiler chickens fed on diets with the addition of EOs, the results showed lower (*p* = 0.02) body temperatures at 12:00 h, than birds without the addition of EOs. Average body temperature (°C) was 39.3 vs. 39.0, respectively for animals on diets with 0 ppm and 200 ppm addition of EOs. Broiler chickens fed on the RED diets, showed lower (*p* = 0.04) body temperatures at 20:00 h, than birds on the STD diets. The average body temperature (°C) was 36.6 vs. 35.4, respectively for animals on the STD and the RED diets. In the total period (1–42 d), the addition of EOs had no effect (*p* ≥ 0.17) on the body temperature of the chickens at the different times of the day. In these evaluations, there was no effect (*p* ≥ 0.40) of the interaction (diet × EO) on the chickens’ body temperature.

### 3.5. Mortality

Mortality was observed only at the finisher phase (21–42 d) in chickens fed on the STD diets, in which three birds died in the group with the STD diet with no addition of EOs (3/50 = 6% of treatment), and two birds in the group with the STD diet with the addition of 200 ppm of EOs (2/50 = 4% of treatment). The RED diets, with or without EO supplementation, did not have any chicken mortality. Table 7 shows the distribution of mortality of birds in the study.

## 4. Discussion

### 4.1. Main Effect: Nutrient Concentration in Diets

In the starter phase, nutrient concentrations in diets were not reflected on the feed intake of broiler chickens. In part, this effect could be related to the adaptation capability of chickens when they are fed on diets with low nutrient concentrations, as broiler chickens try to compensate the low concentration of nutrients by improving the digestion and absorption process. Reduction in energy and protein in diets improves the gastrointestinal tract development as the small intestine has increased villi size, villi:cripta ratio, and mucosa thickness [24,25,26]. In addition to nutrient concentration in diets, the high environmental temperatures in the current study could also influence the productive performance of broiler chickens. Physiological and metabolic modifications of broiler chickens under heat stress are reflected on feed intake reduction as an attempt to alleviate the heat production caused by the digestion of nutrients and metabolism [8]. This adaptation differs from that caused by the nutrient concentrations in diet. Broiler chickens in the starter phase had lower THI than those in the finisher phase (Figure 3), and body temperatures were not influenced by treatments during the starter phase (Table 5 and Table 6). In addition, in this growing phase, temperatures of comfort are higher than those in the finisher phase. According to Cassuce et al. [27], ambient temperatures of 31.3, 25.5, and 21.8 °C are better for chicken growth, for the first, second, and third week of age, respectively. In the present study, ambient temperature had small upper values, particularly during the first 2 weeks.

In the finisher phase or total feeding trial, weight gain or feed intake were greater in broiler chickens fed on the STD diets than those in chickens receiving the RED diets; however, FCR was not different among animals fed on the STD or the RED diets. The greater weight gain was an expected effect in broilers fed on the STD diet; however, the FCR did not improve in those chickens because they also had greater feed intake records. The higher THI and ambient temperatures could have greater influence on productive variables of broiler chickens during the finisher phase. The average body temperatures were higher in broiler chickens fed on STD diets than those recorded in animals fed on the RED diets. In addition, at 20:00 h through the entire study (1–42 d), higher body temperatures were registered in broiler chickens fed on the STD diets. That could be related to the feed intake occurring after 16:00 h, as animals increase the feed intake and also have higher heat loss as observed on higher values of body temperature recordings. On a daily basis, from 12:00 h to 16:00 h, chickens reduced feed consumption as higher ambient temperatures were recorded; that helped the heat production of chickens during this time period of the day. It is well recognized that under heat stress, chickens reduce feed intake to reduce heat production as a result of digestion of nutrients and metabolism [8].

### 4.2. Main Effect: Essential Oils in Diets

In the current study, the addition of EOs (main effect) tended to improve the FCR in the overall feeding period (1 to 42 d). This effect was to be expected as broiler chickens increase productive performance, improvements obtained from the addition of growth promoters are measured in small amounts though [28]. Besides the addition of EOs, weather condition effects must be considered in the productive performance of broiler chickens. The improved productive performance of broiler chickens supplemented with oregano essential oil (OEO) is evident in various reports carried out on comfortable environments where no heat stress is reported. Ruan et al. [29] observed increased feed intake and body weight gain of broiler chickens with dietary OEO supplementation in the diet (150 mg/kg or 300 mg/kg) in comfortable environmental conditions (35 °C for the first week and then decreased by 2 to 3 °C per week to a final temperature of 26 °C). The improvements agree with Zhang et al. [17] using a natural or a synthetic OEO (200 mg/kg OEO and room temperatures staring at 32 °C and reduced by 2 or 3° C weekly until reaching at 22 °C), and with Silva-Vázquez et al. [30] using oregano oils from Mexico (0.40 g of OEO/kg of feed; with an initial room temperature of 34 °C, followed by 32 °C on the second day and over the remainder of the first week, which was then reduced by 3 °C per week until it reached 23 °C). Ding et al. [31] also found improvement of productive performance in broiler chickens when they received synthetic OEO (thymol plus carvacrol) supplementation (200, 400, and 600 mg/kg EOs; with a room temperature of 33 °C in the first week, 28 to 31 °C in the second week, and kept at 24 to 26 °C thereafter). Bosetti et al. [18] observed that microencapsulated essential oils (100, 200, or 400 mg EO/kg of feed, the essential oils consisting of a blend of carvacrol plus cinnamaldehyde) were similar to the addition of antibiotics to diets, but better than the control (no additive) to improve the productive performance of broiler chickens maintained in thermal comfort. In contrast, Botsoglou et al. [32] did not observe enhanced productive performance in broiler chickens when OEO was used (50 mg/kg or 100 mg OEO/kg feed; room temperatures were not declared).

The effect of essential oils from cinnamon (cinnamaldehyde) on productive performance of broilers chickens has been less conclusive. Yang et al. [33] found that encapsulated cinnamaldehyde (100 ppm of diet) and citral (100 ppm of diet), separate or in combination, could improve growth of broiler chickens of nonvaccinated and vaccinated (against coccidiosis) groups to a level comparable to antibiotics, and alter the cecal microbiota composition (room temperature was set at 33 °C on day 0 and then was reduced by 2.5 °C each week). In contrast, Lee et al. [34] did not find a change in productive performance when using thymol, cinnamaldehyde, and a commercial product of cinnamaldehyde in diets for broiler chickens (the ambient temperature was gradually decreased from 32 °C on d 0 to 25 °C on d 21 and was then kept constant). Yang et al. [28] did not observe significant effect of addition of an encapsulated cinnamaldehyde compound on production parameters; they suggested that the lack of effect may be related to the optimum production conditions of chickens in their study.

Results on productive performance of broiler chickens under heat stress have been variable and non-conclusive when diets are supplemented with essential oils. Ghazi et al. [35] observed that the addition of vitamin C or oregano EOs (250 mg of oregano essential oil/kg of diet; alone or combined with vitamin C) improved the weight gain and feed conversion efficiency of broiler chickens in the finisher phase, when ambient temperatures were adjusted to simulate diurnal heat stress (temperature controlled by thermostats; during the first 3 weeks 33 °C was applied, so that the temperature was reduced progressively from 33 to 23.9 °C by the end of the third week). Also, Khan et al. [3] reported improved body weight gain and feed conversion efficiency of broiler chickens under heat stress conditions when they were fed diets with cinnamon powder (CNP) or turmeric powder (TP). Tekce and Gül [36] observed an improvement in productive performance of broiler chickens supplemented with EOs of *Origanum syriacum* (100, 300, or 600 mg EO/kg of diet), under stress-free environments (22 °C) and also under heat-stress conditions (36 °C). In contrast, Vlaicu et al. [37] reported that the addition of oregano EOs did not have an effect on the productive performance of broiler chickens (14–42 days) reared under heat stress (constant, 32 °C and 36% humidity). Also, Gasparino et al. [38] did not observe an improvement in productive performance of broiler chickens supplemented with EOs (capsaicin, carvacrol, cinnamaldehyde, and eugenol) under thermoneutral conditions or in heat stress (32 °C during all feeding period).

In the current study, the addition of EOs tended to improve the FCR of chickens in the overall feeding period (1 to 42 d). The effect of EOs on FCR was more evident in the RED diets than that observed in the STD diets. In the overall feeding period, the addition of EOs improved the FCR of chickens fed on the STD diets, from 1.80 to 1.71, which is a 5.0% improvement, while chickens in the RED diets, an improvement of 9.0% was observed, from 1.89 to 1.72. Broiler chickens fed on the STD diets had lower improvements in the FCR, which could be explained by the higher level of production; as it was previously discussed, when production animals approach their maximum level of production, additives have less of an effect on the improvement of production parameters. In this context, animals receiving diets with lower nutrient concentrations had greater improvements in production parameters by the addition of EOs. In both groups (STD and RED diets), the improvement of FCR could be related to the capability of EOs to alleviate heat stress of chickens. In the current study, the higher body temperatures were recorded at 12:00 h; at this time of the day, the broiler chickens receiving EOs showed lower body temperature records than those chickens without EOs. In a study by Bravo et al. [39], chicken diets supplemented with EOs (100 g of EO/t of feed; carvacrol, cinnamaldehyde, and capsicum oleoresin) improved feed efficiency by 9.8%, and the net energy for production by 12.5%. That study concluded that the EO supplementation reduced the total heat loss compared to the control diet. In another study Ruff et al. [40] found that the addition of EOs (*Lippia origanoides*, *Rosmarinus officinalis*, with beetroot or natural betaine) improved the productive performance of broiler chickens exposed to cyclic heat stress and also reduced their body core temperature when compared to chickens without EO supplementation. During cyclic HS, chickens received 35 °C for 12 h daily from day 7 to day 42, and relative humidity remained constant at 55%.

### 4.3. Carcass Evaluations

The lower carcass weight of broiler chickens fed on the RED diets compared to chickens fed on STD diets was an expected result, that is in accordance with the final body weight of chickens in their respective diets. In those regards, Mallo et al. [24] observed lower body weights and retained energy in tissues of chickens fed on diets with reduced metabolizable energy and amino acids. Johnson et al. [41] also found improvement of productive performance and meat production of broiler chickens fed on diets with higher amino acids concentrations and higher metabolizable energy levels.

In the present study, nutrient concentrations neither influenced the carcass yield nor the breast yield. Chickens fed on RED diets showed greater leg-thigh yield, back yield, and/or wing yields. In contrast, Johnson et al. [41] reported that the RED diets had lower breast yield and had no change in leg yield nor wing yields. Consistently, reductions of crude protein or essential amino acids in diets has reduced breast yield of broiler chickens [42,43]. In the current study, heat stress conditions could contribute to a greater leg-thigh yield or wing yield of broiler chickens. Emami et al. [44] observed greater leg yield or wing yield in broiler chickens reared under heat stress; in that study the effect of greater development of muscles was attributed to the movement of birds, as blood was directed to the legs and wings as a heat dissipation mechanism. This observation is in agreement with Oliveira et al. [45], who also reported greater leg yield in broiler chickens under heat stress conditions.

In this study, carcass traits of broiler chickens were not influenced by the addition of EOs. Similar results have also been observed on various studies (Silva-Vázquez et al. [30] using oregano essential oils from Mexico (0.40 g of OEO/kg of feed); Bosetti et al. [18] using microencapsulated essential oils consisting of a blend of carvacrol plus cinnamaldehyde (100, 200, or 400 mg EO/kg of feed); Alp et al. [46] using oregano essential oils (300 mg OEO/kg of feed); Kirkpinar et al. [47] using oregano essential oil or garlic essential oil at 300 mg/kg of feed, and oregano essential oil at 150 mg/kg of feed and garlic essential oil at 150 mg/feed kg; Amouei et al. [48] using 0.75 g thyme extract /kg of feed; Cristo et al. [49] using a blend of carvacrol, cinnamaldehyde, and eugenol extracted from oregano, cinnamon, and cloves, respectively, at the doses of 100 and 150 g blend extracts/t of feed. In contrast, other studies have reported improvements in carcass traits of broiler chickens when supplemented with EOs. Qaid et al. [13] using cinnamon bark powder at 2, 4, and 6 g/kg feed; Ruff et al. [40] in diets supplemented with 37 ppm EOs of *Lippia origanoides* (LO), or 45 ppm LO and 45 ppm EOs of *Rosmarinus officinalis* (RO) and 300 ppm red beetroot, or with 45 ppm LO, 45 ppm RO, and 300 ppm natural betaine; Khattak et al. [50] using a blend of essential oils from basil, caraway, laurel, lemon, oregano, sage, tea, and thyme at 200, 300, 400, and 500 g/t of feed; Peng et al. [51] using a compound of oregano essential oils at 300 or 600 mg/kg of feed. These differences could be related to differences in the EO preparations (plants, extraction methods, EO combinations) and their concentration, but also to different experimental conditions where the studies have taken place, as well as diet composition. In addition to that, heat stress might contribute to the different responses in carcass traits when chickens are supplemented with EOs.

According to Yilmaz and Gul [52], essential oils exert their effect principally by improving antioxidant activity and decreasing apoptosis and protein degradation, and in chickens under heat stress, that results in enhancements for health, digestion, weight gain, and the carcass characteristics. Nevertheless, the results in carcass characteristics of broiler chickens under heat stress and supplemented with essential oils are not consistent. In broiler chickens under heat stress, an improved carcass yield with EO supplementation was found by Elbaz et al. [53] using 200 mg garlic essential oil (G EO)/kg of feed, or 200 mg lemon essential oil (L EO)/kg of feed, or their mixture (GL EO) at 200 mg/kg diet, respectively for 35 days), and by Ruff et al. [40] using 37 ppm EOs of *Lippia origanoides* (LO), or 45 ppm LO, 45 ppm EOs of *Rosmarinus officinalis* (RO), and 300 ppm red beetroot, or with 45 ppm LO, 45 ppm RO, and 300 ppm natural betaine. In contrast, in broiler chickens under heat stress, null influence of EOs on carcass characteristics was reported by Sariözkan et al. [54] using 300 mg thyme essential oil/kg of feed, which was combined with different vitamins (A, C, and E); Señas-Cuesta et al. [55] using two blends of EOs [EO1: Essential oil of *Lippia origanoides*, thymol chemotype (45 ppm), and herbal betaine (150 ppm). Administration dose: 350 g/Ton food]; [EO2: Essential oil of *Lippia origanoides*, phellandrene chemotype (45 ppm), and herbal betaine (150 ppm). Administration dose: 350 g/Ton food]; and Yilmaz and Gull [56] found only moderate improvements in carcass characteristics of broiler chickens under heat stress with EO addition (200, 400, or 600 mg of cumin EO of seed/kg of feed) conditions.

The pH values are close to neutrality; this is because they were measured before rigor mortis, and, therefore, the lactic acid product of post-mortem anaerobic glycolysis had not been produced [57]. These values, along with those of meat temperature, were not influenced by the concentration of nutrients in the diets or by EO supplementation.

### 4.4. Mortality

Mortality was observed in the finisher phase in chickens receiving the STD diets only; three birds died in the STD diet with no addition of EOs, and two birds in the STD diet with addition of 200 ppm of EOs. Chicken mortality was absent in the RED diets with or without EO supplementation. Though mortality might not be related to EO supplementation, it could be related to the greater body weight of chickens in the STD diets. According to Sosnówka-Czajka and Skomorucha [58], the sudden deaths are more frequent in fast growing chickens in the finisher phase, and the cause of this condition has been related to a heart problem which reduces mortality; they suggest reducing the nutrients in diets and adjusting to the optimal comfort temperature in intensive chicken production. Though more research in these regards is needed, especially considering that in tropical and subtropical areas, the conditions of the summer season lead to heat stress which might influence the performance of broiler chickens.

## 5. Conclusions

In the current study, natural ambient conditions implicated heat stress with higher discomfort in the finisher phase of the broiler chickens. In these climate conditions, productive performance or carcass traits were affected by a reduction in nutrient concentrations in diets. The addition of EOs to diets tended to improve feed conversion efficiency and was more evident in the group on RED diets. In addition, no mortality of broiler chickens was recorded in the groups of RED diets. Body temperature records were better for the group on RED diets with the addition of EOs. Further research is warranted considering greater number of animals, different nutrient levels in diets, and other EOs or feed additives, that could in total improve the production and well-being of broiler chickens under a warm climate.

## Figures and Tables

**Figure 1 animals-14-03179-f001:**
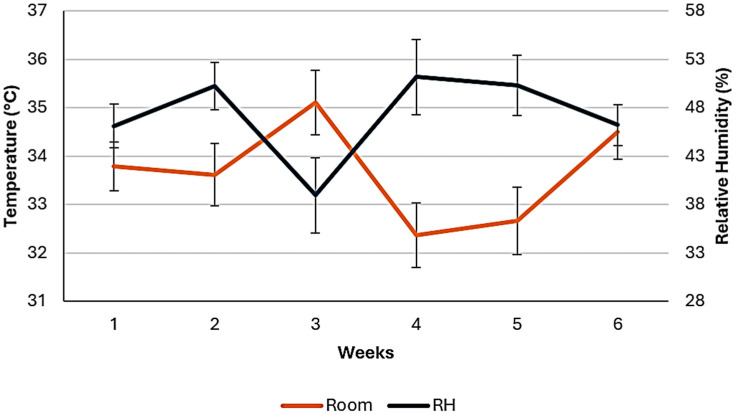
Average values of ambient temperature and relative humidity.

**Figure 2 animals-14-03179-f002:**
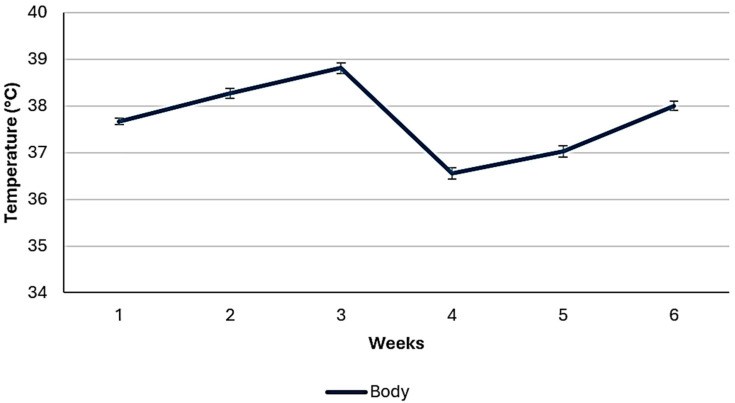
Average body temperatures of broiler chickens.

**Figure 3 animals-14-03179-f003:**
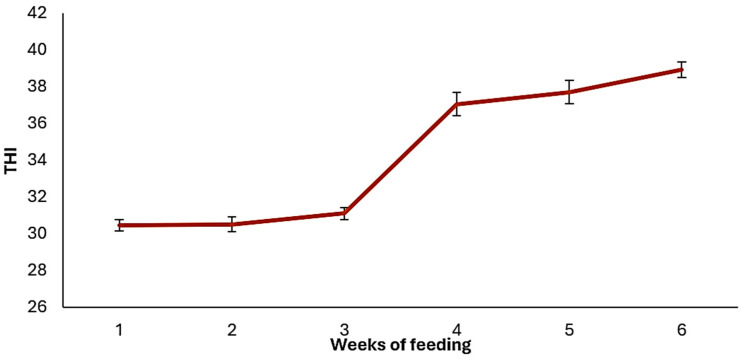
Temperature humidity index (THI) across the study.

**Figure 4 animals-14-03179-f004:**
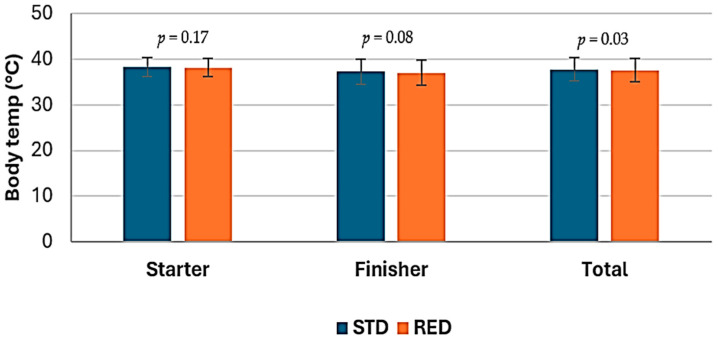
Influence of diet as main effect on body temperature (°C) measured with thermographic camera, by feeding phase of the chickens.

**Figure 5 animals-14-03179-f005:**
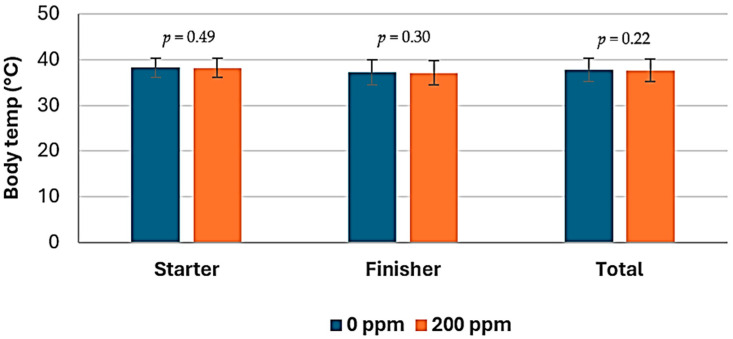
Influence of essential oils as main effect on body temperature (°C) measured with thermographic camera, by feeding phase of the chickens.

**Figure 6 animals-14-03179-f006:**
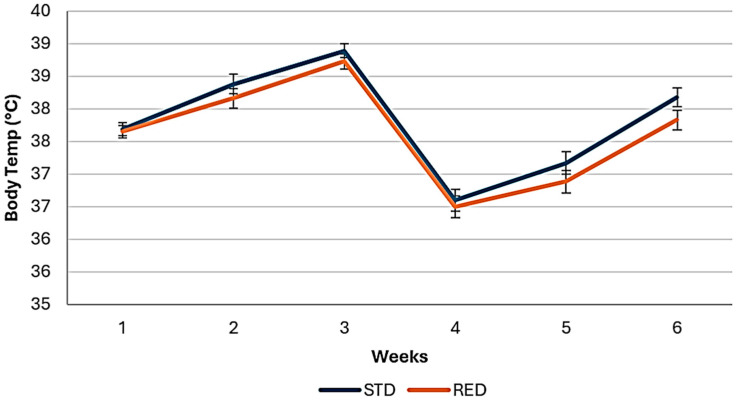
Body temperature, recorded with a thermographic camara, in broiler chickens fed standard (STD) or nutrient reduced (RED) diets, across the study.

**Figure 7 animals-14-03179-f007:**
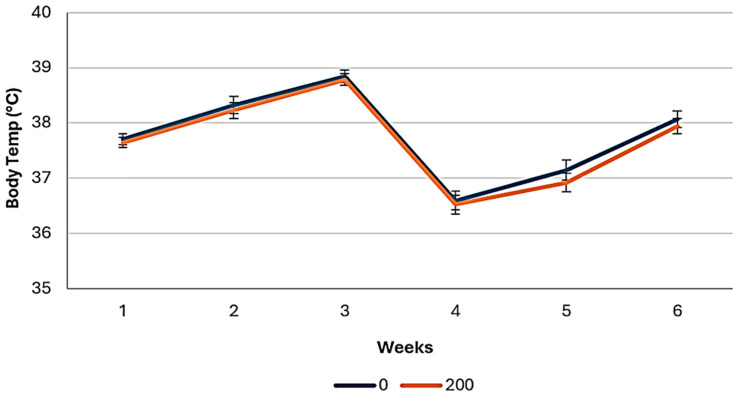
Body temperature, recorded with a thermographic camara, in the broiler chickens fed diets with essential oils (0 ppm and 200 ppm), across the study.

**Table 1 animals-14-03179-t001:** Ingredient and nutrient compositions of the experimental diets (%), for the starter phase (1–21 days of age).

	Standard Diets		Reduced Diets	
Ingredients	0 ppm EOs	200 ppm EOs	0 ppm EOs	200 ppm EOs
Sorghum grain	54.10	54.08	65.20	65.18
Soybean meal	37.40	37.40	30.50	30.50
Vegetable oil (VO)	4.50	4.50	0.30	0.30
Essential oils (EOs)	0.00	0.02	0.00	0.02
Premix *	4.00	4.00	4.00	4.00
Total	100.00	100.00	100.00	100.00
**Nutrient composition**				
Crude protein, %	21.40	21.40	19.20	19.20
Metabolizable energy, kcal/kg	3040	3040	2730	2730
**Nutrient determination**				
Crude protein, %	23.59	23.61	20.76	20.81
Dry matter, %	89.51	89.54	89.01	88.78

Reduced diets had 10% less crude protein and 10% less metabolizable energy than standard diets. Essential oils were obtained from oregano and cinnamon. * Premix: monocalcium phosphate, calcium carbonate, common salt, growth promoter (BDM and 3-nitro), sodium monensin, mineral oil, ethoxyquin, retinol (vitamin A-acetate), cholecalciferol-D3 (vitamin D3), α-tocopheryl acetate (vitamin E), vitamin K3, riboflavin (vitamin B2), cobalamin (vitamin B12), niacin (vitamin B3), calcium D-pantothenate (vitamin B5), choline chloride (vitamin B4), and butylated hydroxytoluene (BHT). Calculated to contain: 21.40% Ca; 8.10% total P; 3.40% Na; 0.80% L-lysine hydrochloride; and 4.15% DL-methionine. The premix for the starter phase of broiler chickens is manufactured and distributed by Trouw Nutrition, Mexico, S.A. de C.V.

**Table 2 animals-14-03179-t002:** Ingredient and nutrient compositions of the experimental diets (%) for the finisher phase (22–42 days of age).

	Standard Diets		Reduced Diets	
Ingredients	0 ppm EOs	200 ppm EOs	0 ppm EOs	200 ppm EOs
Sorghum grain	61.40	61.38	71.00	70.98
Soybean meal	30.10	30.10	24.00	24.00
Vegetable oil (VO)	4.50	4.50	1.00	1.00
Essential oils (EOs)	0.00	0.02	0.00	0.02
Premix *	4.00	4.00	4.00	4.00
Total	100	100	100	100
**Nutrient composition**				
Crude protein, %	18.70	18.70	16.80	16.80
Metabolizable energy, kcal/kg	3120	3120	2890	2890
**Nutrient determination**				
Crude protein, %	19.46	19.58	18.79	18.55
Dry matter, %	89.54	89.43	89.39	89.42

Reduced diets had 10% less crude protein and 10% less metabolizable energy than standard diets. Essential oils were obtained from oregano and cinnamon. * Premix: monocalcium phosphate, calcium carbonate, common salt, growth promoter (BDM and 3-nitro), sodium monensin, mineral oil, ethoxyquin, retinol (vitamin A-acetate), cholecalciferol-D3 (vitamin D3), α-tocopheryl acetate (vitamin E), vitamin K3, riboflavin (vitamin B2), cobalamin (vitamin B12), niacin (vitamin B3), calcium D-pantothenate (vitamin B5), choline chloride (vitamin B4), and butylated hydroxytoluene (BHT). Pre-mix calculated to contain: 19.80% Ca; 3.7% total P; 3.7% Na; 4.3% L-lysine hydrochloride; and 5.2% DL-methionine. The premix for the finisher phase of broiler chickens is manufactured and distributed by Trouw Nutrition, Mexico, S.A. de C.V.

**Table 3 animals-14-03179-t003:** Growth performance of broiler chickens fed diets formulated with different nutrient concentrations and essential oils.

	Standard Diets		Reduced Diets			*p*-Value Main Effect		
	0 ppm EOs	200 ppm EOs	0 ppm EOs	200 ppm EOs	SEM	Diet	EOs	Diet × EOs
Starter phase (1–21 days) ^a^								
Feed intake, g	1351	1323	1283	1246	26.6	0.21	0.56	0.94
Weight gain, g	928	926	770	843	21.7	<0.01	0.31	0.28
FCR	1.46	1.44	1.68	1.48	0.04	0.09	0.14	0.24
Finisher phase (22–42 days)								
Feed intake, g	3443	3213	3039	2959	71.7	0.02	0.24	0.56
Weight gain, g	1750	1736	1522	1601	44.5	0.04	0.71	0.59
FCR	1.99	1.87	2.00	1.85	0.04	0.97	0.14	0.88
Entire feeding period (1–42 days)								
Feed intake, g	4793	4536	4322	4206	86.1	0.02	0.26	0.67
Weight gain, g	2678	2662	2291	2444	62.8	<0.01	0.52	0.43
FCR	1.80	1.71	1.89	1.72	0.04	0.51	0.08	0.56

^a^ Data are means of five replicates of ten animals each. Reduced diets had 10% less crude protein and 10% less metabolizable energy than standard diets. Essential oils (EOs) were obtained from oregano and cinnamon. FCR = feed conversion ratio (g feed/g weight gain).

**Table 4 animals-14-03179-t004:** Carcass characteristics of broiler chickens fed diets formulated with different nutrient concentrations and essential oils.

	Standard Diets		Reduced Diets			*p*-Value Main Effect		
	0 ppm EOs	200 ppm EOs	0 ppm EOs	200 ppm EOs	SEM	Diet	EOs	Diet × EOs
Hot carcass weight, g	3125	3220	2640	2795	81.00	<0.01	0.36	0.82
Carcass yield, % ^a^	78.3	79.2	78.4	78.9	0.37	0.89	0.42	0.81
Breast yield, %	37.0	36.7	34.7	35.0	0.65	0.14	0.99	0.80
Leg-thigh yield, %	26.5	26.5	28.6	27.3	0.30	0.01	0.23	0.22
Back yield, %	20.5	20.9	22.2	21.6	0.28	0.03	0.88	0.33
Wing yield, %	12.0	11.3	13.0	12.6	0.22	<0.01	0.13	0.68
pH ^b^	6.9	6.9	6.8	6.8	0.05	0.36	0.81	0.94
Temperature ^1^	40.7	41.3	39.6	41.0	0.31	0.24	0.11	0.52

^a^ Data are means of five replicates of two animals each. Reduced diets had 10% less crude protein and 10% less metabolizable energy than standard diets. Essential oils (EOs) were obtained from oregano and cinnamon. ^1^ Percentage of carcass yield = (carcass weight, g/live weight, g) × 100. ^b^ pH and Temperature were measured in breast meat after the slaughter process of chickens.

**Table 5 animals-14-03179-t005:** Average body temperature (°C) measured with a thermographic camera, by feeding phase of the broiler chickens in the study.

	Standard Diets		Reduced Diets			*p*-Value Main Effect		
	0 ppm EOs	200 ppm EOs	0 ppm EOs	200 ppm EOs	SEM	Diet	EOs	Diet × EOs
Starter (1–21 d)	38.4	38.3	38.2	38.1	0.05	0.17	0.49	0.98
Finisher (21–42 d)	37.4	37.2	37.1	37.0	0.07	0.08	0.30	0.66
Total (1–42 d)	37.9	37.7	37.7	37.6	0.04	0.03	0.22	0.73

The reduced diets had 10% less crude protein and 10% less metabolizable energy than the standard diets. EOs = essential oils.

**Table 6 animals-14-03179-t006:** Body temperature (°C) measured with a thermographic camera, by feeding phase at different hours of the day.

	Standard Diets		Reduced Diets			*p*-Value Main Effect		
	0 ppm EOs	200 ppm EOs	0 ppm EOs	200 ppm EOs	SEM	Diet	EOs	Diet × EOs
Starter (1–21 d)								
8:00 h	37.6	37.5	37.4	37.3	0.07	0.13	0.64	0.88
12:00 h	39.6	39.5	39.5	39.5	0.12	0.76	0.78	0.90
16:00 h	39.3	39.2	39.2	39.1	0.08	0.59	0.66	0.90
20:00 h	36.9	36.9	36.8	36.7	0.07	0.15	0.53	0.87
Finisher (21–42 d)								
8:00 h	35.1	35.1	34.8	34.9	0.13	0.30	0.98	0.90
12:00 h	39.4	39.1	39.2	38.9	0.07	0.14	0.02	0.80
16:00 h	38.4	38.2	38.3	38.1	0.12	0.53	0.43	0.99
20:00 h	36.5	36.2	35.9	36.1	0.11	0.12	0.78	0.26
Total (1–42 d)								
8:00 h	36.4	36.4	36.1	36.1	36.1	0.16	0.86	0.98
12:00 h	39.5	39.3	39.4	39.2	39.4	0.32	0.17	0.99
16:00 h	38.9	38.7	38.7	38.6	38.7	0.42	0.38	0.94
20:00 h	36.7	36.6	36.4	36.4	36.4	0.04	0.57	0.40

**Table 7 animals-14-03179-t007:** Distribution of mortality of birds in the treatments of the study.

Essential Oils (EOs)	Diet		Total
	STD	RED	
0 ppm EOs	3/50 = 6%	0/50 = 0%	3/100 = 3%
200 ppm EOs	2/50 = 4%	0/50 = 0%	2/100 = 2%
Total	5/100 = 5%	0/100 = 0%	5/200 = 2.5%

## Data Availability

All data appear in the paper. However, additional data that support the findings of this study are available from the corresponding author, Jorge Loredo-Osti [jloredo@docentes.uat.edu.mx], upon reasonable request.

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
