# Peer review of "Productive Performance and Carcass Characteristics of Broiler Chickens Fed on Diets with Different Protein, Energy Levels, and Essential Oils During the Warm Season in Dry Tropics"

_animals, 2024, doi:10.3390/ani14223179_

Round 1

Reviewer 1 Report

Comments and Suggestions for Authors

The experimental design seems too short in terms of both the number of replicates and the total number of animals. Increasing the replicates and the sample size would strengthen the statistical power and reliability of the results.

A thorough review of the English throughout the document is necessary. There are numerous grammatical and structural issues that may affect the clarity and comprehension of the research findings.

Throughout the text, there is a lack of information on the nutritional composition of the diets, which should be determined using wet chemistry methods to ensure accuracy.

Additionally, the results section does not mention if mortality or culling rates were reported, which is crucial for a comprehensive analysis of the study's findings.

When discussing previous studies, it would be helpful to provide more details on their methodologies, such as dosages, durations, and conditions, to help the reader understand the context of the findings.

The section discussing carcass traits influenced by EO would benefit from more specific information on the types and concentrations of essential oils used. Additionally, the potential mechanisms by which EO could affect carcass traits should be elaborated on, especially under varying environmental conditions like heat stress.

The mortality results should be presented in the Results section to provide a clearer and more structured presentation of the findings. Additionally, a statistical analysis of the mortality data is necessary to determine if the differences observed between treatments are statistically significant.

Throughout the text, references to previous studies should be expanded to include more details on methodologies and findings, particularly when discussing the effects of EO and diet modifications.

The Conclusions section should explicitly state the study's limitations and any recommendations for future research to enhance the scientific discussion.

Comments on the Quality of English Language

The document requires an extensive review of the English language. There are multiple instances of grammatical and structural issues that need to be addressed for better clarity and scientific precision.

Author Response

Comments and Suggestions for Authors

  1. The experimental design seems too short in terms of both the number of replicates and the total number of animals. Increasing the replicates and the sample size would strengthen the statistical power and reliability of the results.

 Response: good observation, the increase in the number of animals and replicants in treatments will improve the statistical results. However, our results are adequate considering the low variation in the SEM of treatments. A low SEM value indicates a more accurate estimate of the population mean.

There are articles with the number of replicas used like this work.

http://dx.doi.org/10.1590/1806-9061-2022-1694

https://doi.org/10.3382/ps.2011-01393

http://dx.doi.org/10.1590/1806-9061-2021-1511

  1. A thorough review of the English throughout the document is necessary. There are numerous grammatical and structural issues that may affect the clarity and comprehension of the research findings.

Response: An expert in this field revised the manuscript to improve the English language.

  1. Throughout the text, there is a lack of information on the nutritional composition of the diets, which should be determined using wet chemistry methods to ensure accuracy.

Response: We included this information in table 1 and 2, thanks by the observation. We analyzed for crude protein and dry matter

  1. Additionally, the results section does not mention if mortality or culling rates were reported, which is crucial for a comprehensive analysis of the study's findings.

Response: good observation; we included this information in the results. Also, we included in material and methods a section for mortality.

  1. When discussing previous studies, it would be helpful to provide more details on their methodologies, such as dosages, durations, and conditions, to help the reader understand the context of the findings.

Response: We included this information in the manuscript.

  1. The section discussing carcass traits influenced by EO would benefit from more specific information on the types and concentrations of essential oils used. Additionally, the potential mechanisms by which EO could affect carcass traits should be elaborated on, especially under varying environmental conditions like heat stress.

Response: We added this information to the manuscript, and this section was improved, increasing the discussion, and more references were considered. Thans for the observation.

  1. The mortality results should be presented in the Results section to provide a clearer and more structured presentation of the findings. Additionally, a statistical analysis of the mortality data is necessary to determine if the differences observed between treatments are statistically significant.

Response: We added this information. Mortality is presented in results as descriptive statistics. A table with the distribution of mortality in the treatments was included.

We found that using the software for statistics (SAS), the cero mortality in two treatments does not allow statistical analyses by Fisher's exact test. This is why we included the table with the distribution of mortality in the manuscript. 

  1. Throughout the text, references to previous studies should be expanded to include more details on methodologies and findings, particularly when discussing the effects of EO and diet modifications.

Response: We added this information into the manuscript

  1. The Conclusions section should explicitly state the study's limitations and any recommendations for future research to enhance the scientific discussion.

Response: We added this information into the manuscript, thanks by the observation

  1. Comments on the Quality of English Language

The document requires an extensive review of the English language. There are multiple instances of grammatical and structural issues that need to be addressed for better clarity and scientific precision.

Response: An authority in the field revised the manuscript to improve the English language.

Reviewer 2 Report

Comments and Suggestions for Authors

This study aims to evaluate the productive performance and carcass traits of broiler chickens during the warm season in dry tropics conditions. In my opinion, this paper lacks good preparation and organization, deep scientific explanations, and language revision. So, I recommend improving it, adjusting the previously mentioned issues, and re-submitting this manuscript.

Specific comments:

The title needs to be concise. Change “level” to “levels”

L.11 and L.90: What is the difference between “productive behavior" and “productive performance”?

L.12: Add “chicks” instead of “chickens”.  Correct “on dry tropics conditions” to “in dry tropical conditions”.

L.13: Change “Treatments” to “Chicks or Birds”.

L.14: “0-200 ppm” per what?

L.14: “The RED had 10% less energy and protein, respectively, than STD”: Re-write it clearly.

L.15: “greater”: than what?!

L.15-16: “Productive variables did not have effect (P≥0.14) by EO”: What about the remaining treatments?

Introduction: Needs update and re-organization. Also, there are missed references for some statements, such as in L.34-36 and L.43-46.

L.86-88: I am not in agreement with you!

Materials & Methods: All details must be included.

Statistical analyses: Please rewrite it correctly.

Discussion: Lacks in deep scientific explanations.

L.350-351: What is your explanation? Expand.

L.352-359: Needs careful revision.

Conclusion: Needs to be rewritten and supported with the recommended treatment.

Carefully revise tables 4 and 5, particularly the value significance.

Comments on the Quality of English Language

Acceptable, with some needed improvements.

Author Response

Comments and Suggestions for Authors

This study aims to evaluate the productive performance and carcass traits of broiler chickens during the warm season in dry tropics conditions. In my opinion, this paper lacks good preparation and organization, deep scientific explanations, and language revision. So, I recommend improving it, adjusting the previously mentioned issues, and re-submitting this manuscript.

Response: thank you by the comment, the manuscript was improved considering all the observations made by reviewers and editor.

Specific comments:

  1. The title needs to be concise. Change “level” to “levels”

Response: it was corrected

L.11 and L.90: What is the difference between “productive behavior" and “productive performance”?

Response: throughout the manuscript it is used “productive performance”

L.12: Add “chicks” instead of “chickens”.  Correct “on dry tropics conditions” to “in dry tropical conditions”.

Response: it was corrected

L.13: Change “Treatments” to “Chicks or Birds”.

Response: it was corrected, thanks by the comment.

L.14: “0-200 ppm” per what?

Response: the ppm is correct, it not necessary more explanation. In different case, other units need to be specified, as example mg/kg

L.14: “The RED had 10% less energy and protein, respectively, than STD”: Re-write it clearly.

Response: it was corrected.

L.15: “greater”: than what?!

Response: it was corrected

L.15-16: “Productive variables did not have effect (P≥0.14) by EO”: What about the remaining treatments?

Response: please note that in the previous line was mentioned the effect of diet on weight gain of birds. We complete the sentence mentioning that it is in the started phase. Sometimes it is difficult to express the complete sentences by the limit in words in the abstract. Your comment is adequate, thanks.

Introduction: Needs update and re-organization. Also, there are missed references for some statements, such as in L.34-36 and L.43-46.

Response: we revised this aspect. L.34-36, it is well presented, and all this paragraph is from the reference included at the end of the paragraph. The l.43-46 also is well presented, and it was formed with the two references included at the end of the paragraph.

L.86-88: I am not in agreement with you!

Response: good point. We made a change in the paragraph, thanks. We have been revising this aspect for a time, and still the information is limited, particularly considering the effects of EO, and diets, and related to warm climatic conditions.

Materials & Methods: All details must be included.

Response: We revised this aspect; we improved this section

Statistical analyses: Please rewrite it correctly.

Response: We revised this aspect; we improved this section

Discussion: Lacks in deep scientific explanations.

Response: It was improved significantly. Thanks

L.350-351: What is your explanation? Expand.

Response: it was improved. In other section of manuscript, it is explained the action mode of EO to improve productive performance of chickens.

L.352-359: Needs careful revision.

Response: it was revised and improved

Conclusion: Needs to be rewritten and supported with the recommended treatment.

Response: good observation. Other reviewers suggested to include a recommendation for further research based on the present manuscript: limitations and perspectives.

Carefully revise tables 4 and 5, particularly the value significance.

Response: We revised the tables.

Comments on the Quality of English Language is Acceptable, with some needed improvements.

Response: the manuscript was submitted for English Language revision with an expert in the area.

Reviewer 3 Report

Comments and Suggestions for Authors

The manuscript titled “Productive behavior and carcass characteristics of broiler chickens fed on diets with different protein and energy level and essential oils during the warm season in dry tropics” explores the effects of a blend of cinnamaldehyde and carvacrol essential oils with standard and reduced diets as nutritional manipulation strategy to mitigate heat stress. However, the manuscript is poorly written, inadequately prepared, and raises several major concerns.

-        The experimental diets in Tables 1 and 2 are critically flawed, as they do not meet the recommended macronutrient levels for broilers. The diets were formulated using only sorghum grain, soybean meal, premix, and vegetable oil, with no additional sources of calcium, phosphorus, or essential amino acids. This leads to severe nutrient deficiencies, affecting broiler performance and compromising the reliability of the results. For instance, the standard starter diet in the study contains only 0.13% calcium, 0.1% phosphorus, and 0.29% methionine, which are 87%, 78%, and 50% lower than the recommended levels of these macronutrients, respectively, as per NRC (1944). Given this, the study's findings cannot be accurately adjudicated.

-        The term "Productive behavior" is misleading in the title, as no actual behavioral traits were measured. The study only evaluated growth performance, carcass traits, and body temperature!!!

-        Ambient temperature and relative humidity are not independent variables and cannot be statistically analyzed in relation to the dependent variables. This approach is only applicable to body temperature.

-        In Figure 1, the authors should separate the body temperature results in a different figure from the ambient temperature and relative humidity data.

-        The exact concentrations of both essential oils in the commercial blend must be specified.

-        The study reports that birds were subjected to 24 hours of light daily during the whole trial, which violates the birds' welfare guidelines and likely impacts their productive performance.

-        Line 129, NRC (1994) is an old recommendation for broilers. You have to use the recent nutrient recommendation of the strain.

Comments on the Quality of English Language

The manuscript is poorly written and needs moderate English language editing.

Author Response

Comments and Suggestions for Authors

The manuscript titled “Productive behavior and carcass characteristics of broiler chickens fed on diets with different protein and energy level and essential oils during the warm season in dry tropics” explores the effects of a blend of cinnamaldehyde and carvacrol essential oils with standard and reduced diets as nutritional manipulation strategy to mitigate heat stress. However, the manuscript is poorly written, inadequately prepared, and raises several major concerns.

Response: thank you by the comment. The manuscript has been improved considering all comments made by reviewers and editor.

- The experimental diets in Tables 1 and 2 are critically flawed, as they do not meet the recommended macronutrient levels for broilers. The diets were formulated using only sorghum grain, soybean meal, premix, and vegetable oil, with no additional sources of calcium, phosphorus, or essential amino acids. This leads to severe nutrient deficiencies, affecting broiler performance and compromising the reliability of the results. For instance, the standard starter diet in the study contains only 0.13% calcium, 0.1% phosphorus, and 0.29% methionine, which are 87%, 78%, and 50% lower than the recommended levels of these macronutrients, respectively, as per NRC (1944). Given this, the study's findings cannot be accurately adjudicated.

Response: the comment is important; in fact, we also had the same concern. However, the nutrients in diets are adequate. In the estimation, the supply of nutrients made by the soybean meal and by the premix must be considered. Please note that the premix includes minerals, amino acids, vitamins and other ingredients to ensure the nutrients in the diet. The premix is elaborated by the experts of Trouw Nutrition, Mexico, S.A. de C.V.

- The term "Productive behavior" is misleading in the title, as no actual behavioral traits were measured. The study only evaluated growth performance, carcass traits, and body temperature!!!

Response: good observation, we corrected this term, thanks

- Ambient temperature and relative humidity are not independent variables and cannot be statistically analyzed in relation to the dependent variables. This approach is only applicable to body temperature.

Response: We revised this aspect. Climatic conditions are important for the study. In the statistical analysis only body temperature was considered. To avoid confusion, we deleted the correlation coefficient and only showed the figure with the climatic conditions

- In Figure 1, the authors should separate the body temperature results in a different figure from the ambient temperature and relative humidity data.

Response: We separated these figures.

- The exact concentrations of both essential oils in the commercial blend must be specified.

Response: We would like to include this information, unfortunately the technical note of the commercial blend does not specify these concentrations; we included in material and methos the commercial blend of EO used.

- The study reports that birds were subjected to 24 hours of light daily during the whole trial, which violates the birds' welfare guidelines and likely impacts their productive performance.

Response: good point. In further research we are going to include. Other reports also considered 24 h of light. Thank you for your observation.

https://doi.org/10.1016/j.psj.2023.102825

DOI: 10.9775/kvfd.2012.8474

- Line 129, NRC (1994) is an old recommendation for broilers. You have to use the recent nutrient recommendation of the strain.

Response: We agree with the comment. Previously we made two studies regarding the nutrient concentration in diets for our conditions. The present formulations considered these reports. We have included in the manuscript these two reports. Please note that our formulations are close to the Ross (308) recommendations, but we used 2 phases of feeding, due to practical management. The Ross recommendation is for 3 phases of feeding.

Comments on the Quality of English Language

The manuscript is poorly written and needs moderate English language editing.

Response: the manuscript was submitted for English Language revision with an expert in the area.

Reviewer 4 Report

Comments and Suggestions for Authors

The article is an original scientific study that provides important information. It will make significant contributions to the literature. It is important to be a detailed study. Necessary arrangements must definitely be made.

Author Response

The article is an original scientific study that provides important information. It will make significant contributions to the literature. It is important to be a detailed study. Necessary arrangements must definitely be made.

Response: thank you for the comment

REPORT:

 Title: The title is compatible with the content.

Response: thank you for the comment

Simple Summary:

Response: we have included a simple summary

Abstract: The findings are written in a understandable manner, appropriate to the content of the article.

Response: thank you for the comment

Keywords: It is written in accordance with the content of the study and is understandable.

Response: thank you for the comment

Introduction: The purpose of the study is stated. Appropriate literature support has been provided according to the purpose. Current studies have been evaluated. The study is scientifically significant.

Response: thank you for the comment

Materials and Methods:

The material and method section is well written. Appropriate methods were used. However, the following questions should be answered.

Line 107-109: “The experiment started on June 29 and ended on August 10 of 2023, during this period of time the average temperature and relative humidity were 32.0±1.12 °C and 56.9±5.20 %, respectively. “ Is the temperature and humidity here the outside of the coop? Is it the inside of the coop?

The conditions inside the coop should be given in more detail. Ventilation, fans, windows, lighting, etc.

Response: thank you for the comment, the information was included in the manuscript.

Line 173: At the beginning of the trial, equal numbers of males and females were given. Was this male-female ratio taken into consideration during the slaughtering process? If it has been done, it should be stated.

Response: We considered only male chickens at the slaughtering process; we included the information in the manuscript

Literature support was obtained for the methods.

Response: thank you for the comment

Statistical analysis is appropriate.

Response: thank you for the comment

Results: Results appropriate for the purpose have been obtained. Tables and figures are understandable and scientific. Tables and figures related to the subject have been given. It is written in an understandable language.

- 3.1. Climatic conditions: Body temperatures according to the experimental groups are given in Figure 2, Figure 3 and Table 6. It is not necessary to give body temperatures here.

Response: thank you by the comment. Body temperatures are shown in figure 4 (effect of diet) and figure 5 (effect of EO), and Table 6. We believe that this information is important for the reader, and we find that this place is adequate for the presentation of the information.

Discussion: The discussion section is given in accordance with the study findings. Necessary explanations have been made. It was found appropriate. Reference is made to the literature. However, the following point should be noted. Neither the method nor the findings are given.

- Mortality: There is no information given in the method section regarding mortality. It should be given. Also, the findings should be given in a table or figure. It is not correct to be like this.

Response: good point. We agree with the comment, we are including this information in the manuscript

Conclusions: More clear results should be given in the conclusion section. For example; It seems that this diet is better to apply. Because the dose was tested in this study.

Response: conclusions were modified according to the comments of all reviewers.

Owerall:

The article is an original scientific study that provides important information. It will make significant contributions to the literature. It is important to be a detailed study. Necessary arrangements must definitely be made.

Response: thank you for your comments. The manuscript was improved.

Round 2

Reviewer 1 Report

Comments and Suggestions for Authors

In lines 46-48, the sentence "Broiler chickens are important given their commercial value as well as a nutrient supply for the population" can be simplified for clarity. It is recommended to revise it to: "Broiler chickens are commercially important and provide a key nutrient source for the population."

Throughout lines 46-57, the term "broiler chickens" is repeated frequently. To enhance readability, it would be better to alternate with terms like "birds" or "chickens."

In lines 294-295, the phrase "Tended (P=0.08) to show lower body temperature than those birds receiving the STD diets" is somewhat vague. It is suggested to clarify this by stating: "Birds on RED diets showed a non-significant trend towards lower body temperature compared to those on STD diets (P=0.08)."

In lines 123-129, the description of the ventilation and temperature control could benefit from additional details.

In lines 142-145, the rationale behind choosing 200 ppm of essential oils (EO) should be clarified. Including reasoning based on previous studies or justification for this specific dosage would strengthen the methodology.

In lines 199-201, the process for selecting two chickens per pen for carcass analysis should be clarified in more detail. While it is mentioned that the selection was random, it is important to specify if any criteria, such as body weight or health status, were used to exclude certain animals.

In lines 217-226, the statistical analysis section would benefit from detailing how key assumptions, such as normality and homogeneity of variance, were verified before applying the GLM procedure. If these assumptions are not met, alternative statistical models or transformations may be more appropriate. For instance, non-parametric tests could be applied if normality is violated, or generalized linear models (GLMs) with alternative link functions and distributions (such as Poisson, binomial, or gamma) could be used to better fit the data.

In lines 274-283, there is some inconsistency regarding carcass yield trends. The discussion should clarify whether these differences are statistically significant or simply observed trends to avoid confusion.

Lastly, in lines 291-299, the explanation of body temperature changes across the study could be structured more clearly. Providing a more detailed description, along with improved visual aids, would help readers understand the impact of the diets on temperature regulation in the chickens.

Author Response

  1. In lines 46-48, the sentence "Broiler chickens are important given their commercial value as well as a nutrient supply for the population" can be simplified for clarity. It is recommended to revise it to: "Broiler chickens are commercially important and provide a key nutrient source for the population."

Response: We agree with the comment. We made the change into manuscript.

  1. Throughout lines 46-57, the term "broiler chickens" is repeated frequently. To enhance readability, it would be better to alternate with terms like "birds" or "chickens."

Response: We made the change into manuscript

  1. In lines 294-295, the phrase "Tended (P=0.08) to show lower body temperature than those birds receiving the STD diets" is somewhat vague. It is suggested to clarify this by stating: "Birds on RED diets showed a non-significant trend towards lower body temperature compared to those on STD diets (P=0.08)."

Response: We made the change. “There was no effect of treatment on body temperature of chickens during the starter phase (P≥0.17) neither in the finisher phase (P=0.08).”

  1. In lines 123-129, the description of the ventilation and temperature control could benefit from additional details.

Response: We revised this aspect, and we find that the information is adequate.

  1. In lines 142-145, the rationale behind choosing 200 ppm of essential oils (EO) should be clarified. Including reasoning based on previous studies or justification for this specific dosage would strengthen the methodology.

Response: We agree with the comment. The selection of 200 ppm of EO was based on the technical note of the EO manufacturer, as by previous reports using similar EO. Zhang et al. [26]      Bosetti et al. [29]. We addressed this information into the manuscript.

  1. In lines 199-201, the process for selecting two chickens per pen for carcass analysis should be clarified in more detail. While it is mentioned that the selection was random, it is important to specify if any criteria, such as body weight or health status, were used to exclude certain animals.

Response: The selection of the chickens was made by visual appreciation, considering birds within the average size in each cage. Birds of biggest or smaller sizes were excluded.

  1. In lines 217-226, the statistical analysis section would benefit from detailing how key assumptions, such as normality and homogeneity of variance, were verified before applying the GLM procedure. If these assumptions are not met, alternative statistical models or transformations may be more appropriate. For instance, non-parametric tests could be applied if normality is violated, or generalized linear models (GLMs) with alternative link functions and distributions (such as Poisson, binomial, or gamma) could be used to better fit the data.

Response: The assumptions of normality and homogeneity of variance were made before the GLM procedure. These assumptions are met, and the factorial design (parametric test) is appropriate for the statistical analyses. The tests of normality and homogeneity of variance were included in the text of the manuscript.

“The model assumptions for all variables were verified using the Shapiro-Wilk test for normality (P≥0.18) and the Bartlett test for homogeneity of variances (P≥0.11).”

  1. In lines 274-283, there is some inconsistency regarding carcass yield trends. The discussion should clarify whether these differences are statistically significant or simply observed trends to avoid confusion.

Response: We revised this respect to avoid any confusion. The results of carcass characteristics are well described in the manuscript. Carcass weight (g) differs from carcass yield (%). Therefore, carcass of birds fed the STD diets had greater weight than carcass of chickens fed the RED diets. However, treatments had no effect on carcass yield of birds.

  1. Lastly, in lines 291-299, the explanation of body temperature changes across the study could be structured more clearly. Providing a more detailed description, along with improved visual aids, would help readers understand the impact of the diets on temperature regulation in the chickens

Response: We revised this respect, and we included two figures for the main effects (diets or EO):

Figure 4. Influence of diet as main effect on body temperature (°C) measured with thermographic camera, by feeding phase of the chickens.

Figure 5. Influence of essential oils as main effect on body temperature (°C) measured with thermographic camera, by feeding phase of the chickens.

Reviewer 2 Report

Comments and Suggestions for Authors

The manuscript still needs careful revision. There are some comments that you didn't address! 

First of all, the references' order wasn't correct. For example, you cited "Amouei et al. (47)" in the text, but actually you wrote it (46) in the references' list!! Please carefully re-order them.

L.46-55: The mentioned reference wasn't the correct one! Please compare the percentages mentioned in the text to those mentioned in the provided reference.

In the introduction, clarify the meaning of "warm season in dry tropical climate". It is important for readers.

L.76-77: Needs to be supported with related references, such as https://doi.org/10.14202/vetworld.2024.2044-2052

L.135-136: Just this vaccine?

L.138: What was the final live body weight?

L.142: How did you prepare the concentration of 200 ppm?

L.226: Define the abbreviations "GLM and SAS".

L.232: What does the highlighted point mean?!

L.497-505: Revise it again!

References: You didn't follow the journal format for all references, e.g., in Ref. 52.

Remove the last blank page "21".

Author Response

The manuscript still needs careful revision. There are some comments that you didn't address!

Response: We are doing our best to respond to your comments properly. Thank you for the comment.

First of all, the references' order wasn't correct. For example, you cited "Amouei et al. (47)" in the text, but actually you wrote it (46) in the references' list!! Please carefully re-order them.

Response: good point. We revised all manuscript in this respect, thank you for the observation.

L.46-55: The mentioned reference wasn't the correct one! Please compare the percentages mentioned in the text to those mentioned in the provided reference.

Response: We revised the data in detail. We updated the percentages into the manuscript. 

In the introduction, clarify the meaning of "warm season in dry tropical climate". It is important for readers.

Response: at the end of the paragraph, we added this information “when high ambient temperatures cause heat stress in the birds”

L.76-77: Needs to be supported with related references, such as https://doi.org/10.14202/vetworld.2024.2044-2052

Response: We included the references. Thanks for the observation.

L.135-136: Just this vaccine?

Response: Yes, We used only one vaccine.

L.138: What was the final live body weight?

Response: We included this information

L.142: How did you prepare the concentration of 200 ppm?

Response: the 200 ppm corresponds to 0.2 g EO/kg of feed, i.e. For 100 kg of feed are included 20 g of EO (0.2 × 100 =20). The EO were first mixed with the premix and then, the premix plus the EO were mixed with the other ingredients of the diet according to the respective treatments.

L.226: Define the abbreviations "GLM and SAS".

Response: they were included in MM section

L.232: What does the highlighted point mean?!

Response: it means that it was deleted (r = 0.92); another reviewer suggested the removal. It was only to show that we attended the comment. We corrected this aspect into the manuscript

L.497-505: Revise it again!

Response: We revised this section and did only small corrections. Please note that these references are to show the positive influence of EO on carcass characteristics in another studies.

References: You didn't follow the journal format for all references, e.g., in Ref. 52.

Response: We corrected this aspect.

Remove the last blank page "21".

Response: We corrected this aspect.

Reviewer 3 Report

Comments and Suggestions for Authors

-        It is uncommon in commercial practices to include macronutrients like calcium and phosphorus in premixes. The authors should explicitly clarify in the Materials and Methods section that these nutrients were incorporated into the diets via the premix.

-        In the revised version, the authors reported crude protein values that exceeded the calculated ones by at least 2%. Please provide an explanation for this discrepancy.

-        The methods for CP and DM determination must be clearly outlined in the Materials and Methods section.

-        Please ensure that any language edits made to the revised version are highlighted in a different color, making them easier to track and review.

Comments on the Quality of English Language

-        Please ensure that any language edits made to the revised version are highlighted in a different color, making them easier to track and review.

Author Response

-        It is uncommon in commercial practices to include macronutrients like calcium and phosphorus in premixes. The authors should explicitly clarify in the Materials and Methods section that these nutrients were incorporated into the diets via the premix.

Response: good observation. We included this information in MM.

-        In the revised version, the authors reported crude protein values that exceeded the calculated ones by at least 2%. Please provide an explanation for this discrepancy.

Response: Good point. We mainly attribute this effect to the crude protein contained in the premix.

As observed (Tables 1 and 2), the determined CP values exceeded the calculated ones by at least 2%; this could be due to the CP contained in the premix. In the formulation we did not include CP in the premix because the technical recommendation does not consider CP; however, considering the amino acids and other additives, the premix must contain crude protein. It is recommended to analyze all ingredients of the diet, including the premix.

-        The methods for CP and DM determination must be clearly outlined in the Materials and Methods section.

Response: We included this information in MM

-        Please ensure that any language edits made to the revised version are highlighted in a different color, making them easier to track and review.

 Response: it is a good comment. The changes into manuscript are highlighted in yellow.

Reviewer 4 Report

Comments and Suggestions for Authors

Necessary corrections have been made and it is now suitable for publication.

Author Response

Thanks